# Modulation of Apoptosis by Bovine Gammaherpesvirus 4 Infection in Bovine Endometrial Cells and the Possible Role of LPS in This Process

**DOI:** 10.3390/biology13040249

**Published:** 2024-04-08

**Authors:** Florencia Romeo, Santiago Delgado, Marisol Yavorsky, Lucía Martinez Cuesta, Susana Pereyra, Erika González Altamiranda, Enrique Louge Uriarte, Sandra Pérez, Andrea Verna

**Affiliations:** 1Instituto de Innovación para la Producción Agropecuaria y Desarrollo Sostenible (IPADS, INTA-CONICET), Grupo de Salud Animal RN 226, Balcarce 7620, Argentina; romeo.florencia@inta.gob.ar (F.R.); yavorsky.marisol@inta.gob.ar (M.Y.); pereyra.susana@inta.gob.ar (S.P.); galtamiranda.erika@inta.gob.ar (E.G.A.); elougeuriarte@gmail.com (E.L.U.); 2Facultad de Ciencias Agrarias, Universidad Nacional de Mar del Plata, Mar del Plata 7600, Argentina; delgadosantiago@hotmail.com; 3Centro de Investigación Veterinaria de Tandil (CIVETAN), Universidad Nacional del Centro de la Provincia de Buenos Aires—CONICET, Tandil 7000, Argentina; lmartinez@vet.unicen.edu.ar (L.M.C.); seperez@vet.unicen.edu.ar (S.P.); 4Facultad de Ciencias Veterinarias, CISAPA, Universidad Nacional del Centro de la Provincia de Buenos Aires (UNCPBA), Tandil 7000, Argentina

**Keywords:** bovine herpesvirus type 4, bovine endometrial primary culture cells, apoptosis

## Abstract

**Simple Summary:**

Considering that the bovine uterine infections caused by bacteria make the host more susceptible to secondary infections, we investigated the role of BoGHV4 and bacterial LPS in cell death in an experimental model of endometrial cells. Various techniques, including staining and molecular analysis, were used. The results showed that the level of cell death following BoGHV4 infection is linked to the timing of viral infection and the presence of LPS. Apoptosis was most prominent in cells infected with BoGHV4 and BoGHV4 + LPS, which correlated with the occurrence of cytopathic effects and high viral titers. Morphological, biochemical, and molecular changes were observed during both the early and late stages of apoptosis. These discoveries offer valuable insights into the factors influencing BoGHV4 pathogenesis and provide an understanding of the mechanisms underlying viral infection.

**Abstract:**

The prevalent pathogens associated with bovine uterine infections are bacteria that appear to increase the host’s susceptibility to secondary infections with other bacteria or viruses, among which BoGHV4 is the most frequently found. In this work, the study of the pathways of apoptosis induction was carried out on an experimental model of primary culture of endometrial cells, in order to know the implication of BoGHV4 and the presence of bacterial LPS in the pathogenesis of the bovine reproductive tract. For this, different staining techniques and molecular analysis by RT-PCR were used. The results obtained allowed us to conclude that the level of cell death observed in the proposed primary culture is directly related to the time of viral infection and the presence of LPS in BoGHV4 infection. The apoptosis indices in cells infected with BoGHV4 and BoGHV4 + LPS revealed a maximum that correlated with the appearance of cytopathic effects and the maximum viral titers in the model studied. However, morphological, biochemical, and molecular changes were evident during both early and late stages of apoptosis. These findings provide information on the factors that may influence the pathogenesis of BoGHV4 and help to better understand the mechanisms involved in virus infection.

## 1. Introduction

Apoptosis is a genetically regulated mode of programmed cell death (PCD) with the aim of eliminating surplus, damaged, or mutated cells; it plays a key role in the innate response to viral infection [1]. Although it contributes to the prevention of pathogenesis, it often results in a potentially costly sacrifice for the cell [2,3].

Several viruses have evolved strategies to evade apoptosis, allowing them to effectively establish either a lytic infection or a long-term latent infection [4,5,6,7]. It seems that promoting apoptosis appears to offer advantages to herpesviruses during the later stages of their replication cycle, while preventing apoptosis seems to play a crucial role in the transition to latency [8]. The classification of the *Herpesviridae* family into three subfamilies (α, β, and γ) is determined by biological characteristics, including factors such as host range and the length of the infectious cycle. The members of each herpesvirus subfamily are recognized for possessing genes that encode anti-apoptotic functions [9]. Bovine gammaherpesvirus 4 (BoGHV4) is associated with reproductive tract infections occurring in the postpartum period [10], and it has been suggested to play a role in the development of non-responsive postpartum metritis. However, its intrinsic pathogenic power seems low and the prevalence of viral metritis remains unclear [11]. Symptoms develop only when BoGHV4 cooperates with Gram-negative bacteria within the uterus, being currently considered as a cofactor for the development of an inflammatory reaction initiated by bacteria [10]. In vitro experiments demonstrated that tumor necrosis factor alpha (TNF-α), produced by macrophages stimulated by lipopolysaccharides (LPS), induces the expression of the immediate early gene IE2 of BoGHV4, stimulating its replication. This would generate the reactivation of the latent virus in persistently infected animals against a concomitant infection with bacteria [12]. BoGHV4 mainly infects blood lymphocytes (B and T-cells) and shows a particular tropism for the endometrium, in which it causes the death of epithelial and stromal cells [13,14,15]. Although the virus contains anti-apoptotic genes (v-Bcl-2 and v-Flip) in its genome, it appears to trigger apoptosis-mediated cell death. This process is linked to a decrease in Bcl-2 expression, facilitating the completion of BoGHV4’s productive infection cycle [16]. Epithelial cells constitute the initial line of contact with external microbes, making them crucial in safeguarding the host against microbial invasion [17]. As the relationship between BoGHV4 infection and uterine response remains poorly understood, there is significant interest in examining the mechanisms responsible for regulating programmed cell death.

Morán et al. [8] demonstrated variations in the sequence of the v-Flip and v-Bcl2 antiapoptotic genes in Argentine strains of BoGHV4, allowing us to infer that the different genetic characteristics of these genes could be associated with the viral genotype. Furthermore, it is interesting to highlight that the results of different studies carried out on the Madin–Darby bovine kidney (MDBK) cell line suggest that BoGHV4 can produce apoptosis through processes such as the induction of oxidative stress [18].

Our group previously showed that BoGHV4 has a replication preference for BEC cells (bovine endometrial cells) over the MDBK cell line [19]. Moreover, in BEC primary cultures, this virus induces apoptotic cell death, culminating in the release of the virus during the final stage. Currently, elevated viral titers are associated with apoptosis, although the predominant mechanism does not appear to be the mitochondrial pathway [19]. The observed apoptotic responses to BoGHV4 in the previously studied cell lines do not adequately elucidate the in vivo processes. Therefore, it is appropriate to investigate these phenomena in primary culture cells.

Considering that postpartum infection of the bovine uterine tract with LPS-containing pathogens activates lytic replication of BoGHV4 in persistently infected macrophages [20], in this study, we investigated the modulation of apoptosis by BoGHV4 infection in primary cultures of BEC cells and the role of LPS in this process.

## 2. Materials and Methods

### 2.1. Isolation of BEC

The primary culture-derived BEC cells and the fetal bovine serum (FBS) utilized for their cultivation underwent testing through virus isolation in MDBK cells and antigen detection by direct immunofluorescence (DIF) using FITC-labeled polyclonal antibodies (IBR/BHV-1, CJ-F-IBR-10ML; BVDV, CJ-F-BVD-10ML, VMRD, Pullman, WA), as previously described [21]. This test, along with nucleic acid detection using PCR or nested RT-PCR [21,22], was performed to eliminate the possibility of contamination in the initial material with bovine alphaherpesvirus 1 (BoHV-1), BoGHV4, and bovine viral diarrhea virus (BVDV). The BEC cells were incubated in a humidified environment at 37 °C with 5% CO_2_, and the culture medium was refreshed every 48 h until the cells reached confluence.

### 2.2. BoGHV4 Strain

The in vitro experiments utilized the BoGHV4 field strain 07/435, which was initially derived from the vaginal discharge of an aborted cow [22,23]. The virus stock was generated by cultivating the BoGHV4 strain in confluent monolayers of MDBK cells in T-25 flasks (Greiner BioOne, Frickenhausen, Germany) for 48 h. The cells were initially seeded at a density of 1 × 10^5^ cells/mL. Subsequently, the supernatants were collected and stored at −80 °C. Virus titers were assessed using the endpoint titration method on MDBK cells, employing 96-well microtiter plates (Greiner Bio-One, Frickenhausen, Germany). The results were presented as log10 TCID50/mL [24].

### 2.3. Infection of Cells Cultures with BoGHV4 and LPS

BEC cultures at the third passage were cultivated in triplicate within 6-well plates (Greiner Bio-One, Frickenhausen, Germany) at a concentration of 7 × 10^5^ cells/mL. The cultures were maintained at 37 °C in a humidified incubator with a 5% CO_2_ atmosphere. Before conducting the cell assays, confluent monolayers were infected with strain 07/435 at a multiplicity of infection (MOI) of 0.5. Following a 2-hour incubation period, the supernatant was removed and fresh medium (MEM-E with 10% FBS) was added, followed by LPS challenge (100 µg/mL) (LPS from Escherichia coli O55:B5 L6529; Sigma-Aldrich, Burlington, MA, USA). Mock-infected and LPS-treated cells were used as negative controls.

### 2.4. Morphological Analysis by Staining with Rhodamine 123/Propidium Iodide

After 12, 24, and 48 h post infection (hpi), the supernatant was removed by centrifugation at 200 g for 10 min at 4 °C. The pellet was then resuspended in 200 µL/tube of Rod-123 (1 g/mL, prepared in 0.1% PBS-BSA from a 1 mg/mL stock in absolute EtOH). After incubating for 10 min at 37 °C, the cells were washed with PBS and centrifuged again at 200 g for 10 min at 4 °C, and the supernatant was discarded. The pellet was then resuspended in 200 µL of PI (1/100 in PBS-BSA). Approximately 10 µL of the suspension was placed on slides for epifluorescence microscope observation (BX51, OLYMPUS, Tokyo, Japan) using a wavelength of 507 nm.

### 2.5. Detection of Nuclear Morphological Changes by DAPI Staining

BEC cell cultures were cultivated on 12 mm coverslips in 6-well plates (Greiner Bio-one, Frickenhause, Germany) at a density of 3 × 10^5^ cells/mL. Subsequently, the cells were exposed to the BoGHV4 strain 07/435 at a MOI of 0.5. The virus was allowed to absorb for a period of 2 h. After incubation for 2 h, the supernatant was discarded and replaced with fresh medium (MEM-E with 10% FBS) followed by the LPS challenge (100 µg/mL) for 12, 24, and 48 h. Mock-infected and LPS-treated cells were used as negative controls. The fusarium mycotoxin deoxynivalenol (DON, Sigma Aldrich, Buenos Aires, Argentina) was used as the positive control; 500 μL at a concentration of 2.8 μg/mL [25] was used to treat cell monolayers in order to check for the characteristic morphological changes of apoptosis. Following each incubation interval, the culture medium was removed, and cell monolayers were fixed using 4% paraformaldehyde. To assess alterations in nuclear structure, staining with 4′6-diamino-2-phenylindole (DAPI, 1 μg/mL) was performed. The washing procedure using PBS and the dye incubation were conducted in accordance with the supplier’s guidelines (Thermo Scientific™, Waltham, MA, USA). Following incubation, cell monolayers on coverslips were rinsed twice with distilled water and subsequently stained for examination under an epifluorescence microscope (BX51, Olympus, Tokyo, Japan) using a wavelength of 395 nm, as detailed previously [19].

### 2.6. Detection of DNA Fragmentation by TUNEL Assay

The cultivation of BEC cell cultures on 12 mm coverslips within 6-well plates (Greiner Bio-one, Frickenhausen, Germany) was performed at a density of 3 × 10^5^ cells/mL. Subsequently, the cultures were infected with the BoGHV4 strain 07/435 at a MOI of 0.5. Following a 2-h incubation period, the existing supernatant was removed and substituted with fresh medium (MEM-E with 10% FBS), and the cells were exposed to an LPS challenge (100 µg/mL) for 12, 24, and 48 h. Mock-infected and LPS-treated cells were used as negative controls. After each incubation period concluded, the culture medium was removed, and the cells were subsequently fixed using 4% paraformaldehyde. To assess alterations in the structure of cell nuclei, staining with a commercial TUNEL kit (DeadEnd^TM^ Colorimetric TUNEL System, Promega Corporation, Madison, WI, USA) was used according to the manufacturer’s instructions.

### 2.7. Total RNA Isolation, cDNA Preparation, and mRNA Expression Analysis by RT-qPCR

The mRNA expression levels of BcL-2, Bax, and Caspase 3 genes in cell cultures were determined by RT-qPCR. Cells, both infected and uninfected, were collected at 12, 24, 48, and 72 hpi. Subsequently, they were preserved in BIO-ZOL Reagent (PB-L, Argentina) at 80 °C, following the manufacturer’s guidelines, for subsequent RNA extraction. iScript™ (Bio-Rad Laboratories, Inc., Philadelphia, PA, USA) was employed to conduct first-strand cDNA synthesis using an average of 1 μg of total RNA. The cDNA was preserved at −80 °C until the execution of qPCR. The internal control for RT-qPCR was selected as the bovine glyceraldehyde-3-phosphate dehydrogenase (GAPDH) gene. The measurement of its expression level was conducted in accordance with the methodology described by Romeo et al. in 2022 [19]. Specific primers for the BcL-2, Bax, Caspase 3 and GAPDH genes were used [26]. All samples were amplified in triplicate and the qPCR products were expressed as cycle threshold (Ct) values using the software StepOnePlus ^®^, version 2.0) (Applied Biosystems, Foster City, CA, USA). The qPCR analysis utilized the FastStart Universal SYBR Green Master Mix (Rox) from Roche Diagnostics Deutschland, according to the manufacturer’s instructions. The Real-Time PCR CFX96 Touch system from Bio-Rad Laboratories was employed for the procedure. The amplification was conducted under the following conditions: 10 min at 95 °C, 40 cycles of 15 s at 95 °C, and 60 s at 60 °C.

### 2.8. Data Analysis

With the objective of measuring the loss of mitochondrial permeability indirectly expressed as the percentage of Rod-123+ cells (% Rod+), an experiment was carried out using a randomized complete block design with three repetitions and with a factorial arrangement. One of the factors was time with three levels (6, 12, and 24 hpi) and the other factor was cell lines with four levels (Negative Control, LPS, BoGHV4, and BoGHV4+LPS).

A second experiment was conducted to determine the presence of nuclear alterations (DAPI) and detection of in situ DNA fragmentation (TUNEL) were expressed as RAI (Relative Apoptosis Index), using a randomized complete block design with three repetitions and a factorial arrangement. One of the factors was time, with three levels (12, 24, and 48 hpi) and the other factor was cell lines, with four levels (Negative Control, LPS, BoGHV4, and BoGHV4+LPS).

A third experiment was conducted to obtain the relative expression ratios, using a design in randomized complete blocks with a factorial arrangement and with three repetitions. One of the factors was time, with three levels (12, 24, and 48 hpi) and the other factor was the Bax, Bcl2 genes and the relationship between them. The results for the mitochondrial proteins and caspase 3 gene expression were normalized against the expression of the endogenous gene, GAPDH, which had showed no significant differences in the gene expression between untreated and treatment groups. For qPCR results, the Relative Expression Software Tool (REST^®^ v2.0.13, Qiagen Inc., Germantown, MD, USA) was used.

The data were analyzed by analysis of variance followed by mean comparison using the least significant difference test. For all hypothesis tests performed, a significance level of 5% (α = 0.05) was considered. The analysis was carried out through the R program (2023). The values shown in graphs are presented as the mean ± standard deviation (SD) of three independent experiments each done in triplicate. GraphPad Prism 8.0 was used for data plotting for all experiments.

## 3. Results

### 3.1. Evaluation of Early Apoptosis Using Rhodamine 123/Propidium Iodide

The assessment of mitochondrial membrane integrity as an early apoptosis marker through staining with Rod-123 demonstrated a significant interaction (*p* < 0.05) between time and treatment for the last two evaluated time points. As shown in Figure 1, after 12 hpi, the percentage of Rod-123+ cells significantly decreased (*p* < 0.05) in the culture infected with BoGHV4 compared to the uninfected and LPS-treated BEC controls. Additionally, in the culture with BoGHV4+LPS, the decrease in membrane integrity was significant (*p* < 0.05) compared to the culture only infected with the virus. Lastly, significant variations (*p* < 0.05) were observed among the different treatments at 24 hpi, with a decrease in the percentage of Rod-123+ cells in LPS-treated cells compared to the control, in virus-infected cells compared to LPS-treated cells, and in cells doubly exposed compared to those only infected with BoGHV4.

### 3.2. Nuclear Morphology Changes in BEC Cells

DAPI staining of BEC cells infected with the 07/435 strain of BoGHV4 revealed changes in nuclear morphology consistent with those reported in the literature [27], characteristic of late apoptosis, such as chromatin condensation and fragmentation (Figure 2). The results of the RAI analysis, obtained from the count of cells with nuclear alterations, demonstrated a significant interaction between time and different treatments (Figure 3). The cell cultures exhibited cell rounding and reduction in cellular volume along with significant nuclear contraction as early as 12 hpi. This trend continued over time, reaching a significantly higher RAI (*p* < 0.05) at 48 hpi (Figure 3) in both BoGHV4 and BoGHV4+LPS-exposed cells. For the remaining evaluated time points, there were no significant differences (*p* < 0.05) detected among the different treatments.

### 3.3. TUNEL

The cell count with alterations compatible with apoptosis, following TUNEL staining (Figure 4), allowed for the calculation of the RAI in cells exposed to different treatments relative to the negative control. The results demonstrated significant changes (*p* < 0.05) at 24 and 48 hpi in cells only infected with BoGHV4 and BoGHV4+LPS compared to cells only treated with LPS. Furthermore, the RAI values obtained at 48 hpi were 20 times higher than those obtained at 24 hpi, both in cells infected with BoGHV4 and in cells with BoGHV4+LPS (Figure 5).

### 3.4. Pro/Anti-Apoptotic Potential of BoGHV4: Relative Expression of Bax, Bcl-2, and Caspase 3

The pro/anti-apoptotic capacity of the virus was evaluated through the mRNA expression of the mitochondrial proteins Bax and Bcl-2 in BEC cells treated only with BoGHV4. The results of RT-qPCR revealed that the expression levels of the pro-apoptotic gene Bax were similar at 12 hpi and 24 hpi, with a significant increase (*p* < 0.05) at 48 hpi (Figure 6). There was a reduction in the expression levels of the anti-apoptotic gene Bcl-2 at 12 hpi and 24 hpi. However, its expression level increased significantly (*p* < 0.05), reaching its peak at 48 hpi compared to 12 hpi. The analysis of the Bax/Bcl-2 ratio showed no significant differences at the evaluated post-infection times (*p* > 0.05). However, at 12 hpi, a Bax/Bcl-2 ratio of <1 suggests the existence of an anti-apoptotic stimulus, whereas at 24 hpi and 48 hpi, a ratio close to 1 indicates an equilibrium between pro- and anti-apoptotic genes.

When the final stage of apoptosis was evaluated, significant differences (*p* < 0.05) were observed at all evaluated time points for the expression of Caspase 3 mRNA in CEB cells after infection with BoGHV4 and/or treatment with LPS (Figure 7). A significant decrease in Caspase 3 expression was observed at 12 hpi (*p* < 0.05) in all treatments compared to the control. After 24 hpi, Caspase 3 expression increased significantly (*p* < 0.05) in cells infected with BoGHV4 (134 times higher), while in the presence of both the virus and LPS (BoGHV4+LPS), the opposite effect was observed, showing a significant decrease (*p* < 0.05). After 48 hpi, there were only significant changes in cells exposed to LPS, with an increase in the expression of the studied gene, while at 72 hpi, significant increases (*p* < 0.05) were observed in all treatments compared to the control, where Caspase 3 expression was 131 times higher in cells infected with BoGHV4 (Figure 7).

## 4. Discussion

Apoptosis is a complex process that involves a wide variety of biochemical and morphological changes that lead to the complete disappearance of the cell without inducing an inflammatory response [28]. Many viruses modulate this PCD as a strategy to safeguard their replication and survival in infected cells [29].

In this work, diverse experiments were carried out in order to relate the pathogenicity of the Argentine strain 07/435 of BoGHV4 with the apoptosis evidenced in vitro in primary culture of BEC cells and the role of LPS in this process. For the analysis of the pro- and anti-apoptotic potential of the virus, different techniques were used to detect early and late apoptotic events at different times post infection.

The change in the mitochondrial membrane permeability is a phenomenon that often precedes all the other modifications inherent to apoptosis. The results showed an alteration of the membrane potential in the presence of BoGHV4 and BoGHV4+LPS starting at 12 hpi, which decreases considerably at 24 hpi, indicating that mitochondrial damage has already occurred at this time post-infection. It is notable, at this last time point evaluated, that the effect of BoGHV4+LPS is cumulative, where the loss of mitochondrial membrane permeability is greater compared to treatment with BoGHV4 alone or LPS alone. This finding suggests that the presence of bacterial LPS predisposes cells to a greater mitochondrial alteration to subsequent BoGHV4 infection.

Then, changes in nuclear morphology, which could suggest early apoptosis, were assessed through DAPI staining. Significant changes were recorded after 48 hpi in cells infected with the virus and in cells infected and treated with LPS. These results would indicate that the cells are undergoing an apoptotic process. However, some authors mention that morphological changes are not always specific indicators of cell death by apoptosis and may vary between different cell lines [30]. For that reason, they recommend complementing the analysis of this parameter with other methodology, such as the study of DNA fragmentation. Consequently, we proceeded to determine the DNA fragmentation comparing the RAIs of BEC cells after different treatments using TUNEL. Changes in nuclear morphology can be initiated as early as 30 min to 4 h after apoptotic stimulation, and precede changes in the cytoskeleton. In contrast, DNA fragmentation occurs later (24 hpi); hence, TUNEL staining is a good indicator of the late apoptosis process [31,32,33]. Therefore, DAPI and TUNEL techniques were used in a complementary way as indicators of early and late cell damage processes, respectively. TUNEL staining indicated significant apoptosis starting at 24 hpi in cells infected with BoGHV4 and in the presence of BoGHV4+LPS, while at 48 hpi the apoptosis indices following the aforementioned treatments were higher. These results correlate with the observed cytopathic effects, which became evident at 24 hpi and was marked after 48 hpi, indicating a large percentage of dead cells. In turn, it is consistent with the viral titers obtained in previous works [22], which means that at 48 hpi, cell death by apoptosis would allow the release of progeny, thus increasing the extracellular viral load.

The analysis of the mRNA expression of mitochondrial Bcl2, Bax, and the Bax/Bcl-2 ratio in BoGHV4-infected BEC cells demonstrated a “balance”. These findings can be interpreted as a tactic of the virus to promote latency during the initial infection phase, preventing the apoptosis of the infected cells. These results are consistent with those reported in the literature [13], where BoGHV4 efficiently infected purified populations of bovine endometrial epithelial and stromal cells, leading to non-apoptotic cell death and de novo viral production. Finally, the expression of Caspase 3 was evaluated, since it is a central enzyme in the apoptosis process that plays a fundamental role in the activation and execution of the apoptotic cascade. The results obtained from the expression of Caspase 3 were variable according to the different times evaluated. However, it can be noted that LPS by itself is not a potent inducer of Caspase 3 expression in the BEC cells evaluated. On the other hand, it is evident that BoGHV4 can induce a significant increase in Caspase 3 mRNA levels after 24 hpi and 72 hpi and this effect seems to be inhibited in the presence of bacterial LPS. It is important to highlight that these results were obtained in the context of a specific study using BEC cells in particular. Results may vary depending on cell type and experimental conditions used. For example, Sharifi et al. [34], who studied the effect of LPS on neuronal cell apoptosis, demonstrated a significant increase in the expression of caspase 3 together with the pro-apoptotic protein Bax.

Although only Caspase 3 gene expression was evaluated, the results confirm that changes occur at the level of mRNA expression in this cell type. These increases could suggest that BoGHV4 stimulates apoptosis through the caspase pathway, but not in the presence of LPS, where it would use another pathway or would not be the main one.

The results obtained allow us to confirm that BoGHV4 infection results in morphological, biochemical, and molecular changes during the early and late stages of programmed cell death. Furthermore, the magnitude of cell death observed in the proposed primary culture is directly related to the length of viral infection, and to the presence of LPS in BoGHV4 infection. Similar results were observed by Pagnini et al. [18] in MDBK cells, where they demonstrated an inoculum- and infection time- dependent induction of apoptosis, where antioxidants prevented it but did not affect viral replication, confirming that apoptosis is not essential in productive BoGHV4 infection. In contrast, Morán et al. [8] demonstrated the variability in the ability to induce changes compatible with apoptosis of phylogenetically divergent BoGHV4 strains, and the variations present according to the type of infected cell. This is similar to what was found by other authors regarding the time of presentation of apoptosis indicator characteristics, such as morphological changes or nuclear condensation, who point out that there are variations according to different cell types [30,35]. Furthermore, it is important to consider that some stimuli can cause cell death by apoptosis or necrosis dependent on their intensity and duration, as well as the cell type on which they act [33].

## 5. Conclusions

The data presented in this study showed that the degree of apoptosis in the proposed primary culture is directly related to the length of viral infection and to the presence of LPS in BoGHV4 infection. This finding provides insight into the factors that may influence the pathogenesis of BoGHV4 and helps to achieve a better understanding of the mechanisms involved in viral infection, as well as to identify the viral genes involved in the control of apoptosis by other gammaherpesviruses. Understanding the antiapoptotic mechanisms of bovine herpesvirus will greatly enhance the ability to develop new antiviral compounds and vaccines for treatment and prevention. These new antiviral compounds could inhibit virus release, adjust latency to reduce viral infection, or promote death of infected cells in early infection.

## Figures and Tables

**Figure 1 biology-13-00249-f001:**
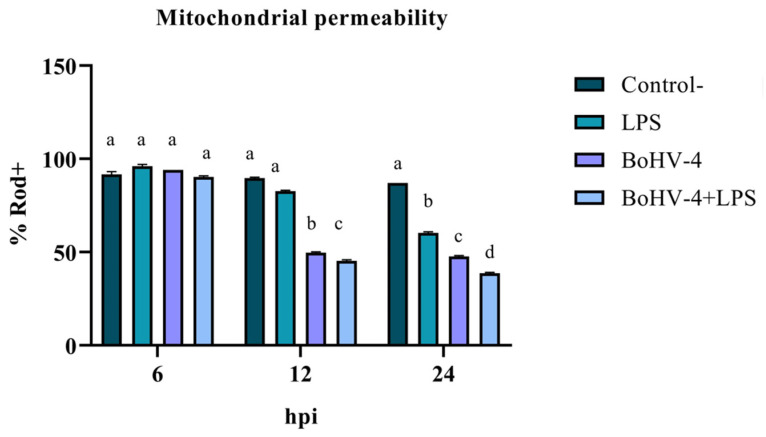
Percentage of Rod-123+ BEC cells at different times post infection with BoGHV4 and/or treatment with LPS. Means accompanied by the same letter indicate non-significant differences (α = 0.05) between the treatments at the evaluated times.

**Figure 2 biology-13-00249-f002:**
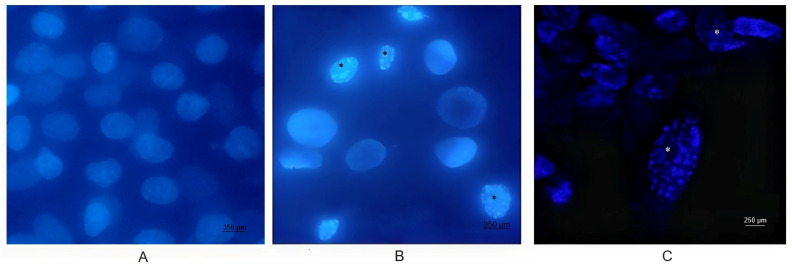
Conventional ((**A**). Control, (**B**). Infected BEC cells) and confocal ((**C**). Infected BEC Cells) fluorescence microscopy of DAPI-stained BEC cells infected with BoGHV4. Nuclear morphological changes characteristic of apoptosis (asterisks) were observed at a magnification of 40×.

**Figure 3 biology-13-00249-f003:**
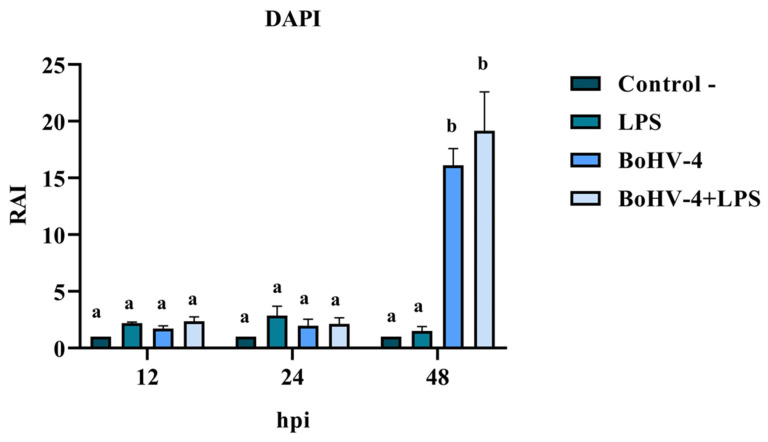
Means and standard deviations of Relative Apoptosis Index (RAI), determined after infection with BoGHV4 and/or LPS treatment at 12, 24, and 48 hpi in BEC cells (DAPI staining). Means accompanied by the same letter indicate non-significant differences (α = 0.05) between each treatment at reading time.

**Figure 4 biology-13-00249-f004:**
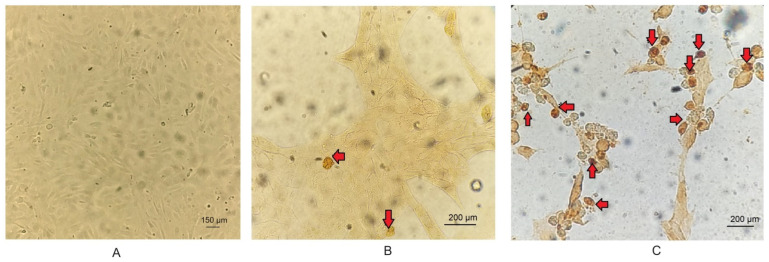
Representative images of apoptotic cells detected by TUNEL (arrows) in Control BEC cells (**A**) BoGHV4-infected BEC cells after 24 (**B**) and 48 (**C**) hpi. 2× magnification.

**Figure 5 biology-13-00249-f005:**
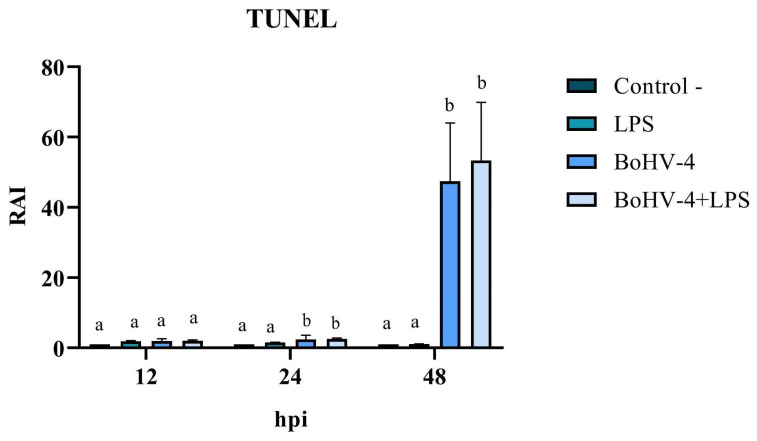
Relative Apoptosis Index (RAI) in BEC cells infected with BoGHV4 and/or treated with LPS. Detection through TUNEL. Means accompanied by the same letter indicate non-significant differences (α = 0.05) between the treatments at the evaluated times.

**Figure 6 biology-13-00249-f006:**
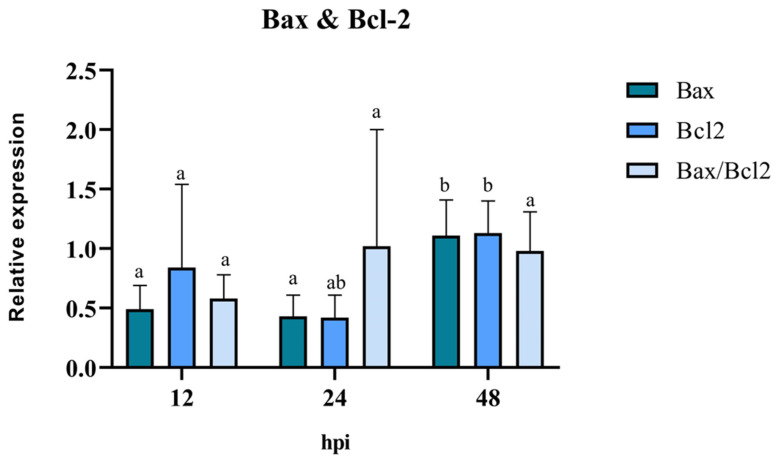
Means and standard deviations of the relative expression of Bax, Bcl-2, and Bax/Bcl-2 in BEC cells at different times post infection with BoGHV4. The data were normalized to the expression of the reference gene GAPDH and are presented in arbitrary units. Levels of times for each gene with the same accompanying letter indicate that there are non-significant differences (α = 0.05) between them.

**Figure 7 biology-13-00249-f007:**
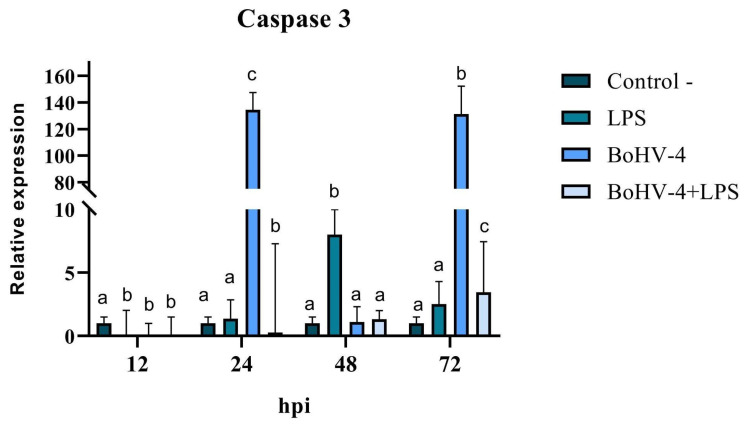
Relative expression of Caspase 3 in BEC cells after different times post-infection with BoGHV4 and/or treatment with LPS. The data were normalized to the expression of the reference gene GAPDH and are presented in arbitrary units. Means accompanied by the same letter indicate non-significant differences (α = 0.05) between the treatments at the evaluated times.

## Data Availability

Data are contained within the article.

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
