# Peer review of "Modulation of Apoptosis by Bovine Gammaherpesvirus 4 Infection in Bovine Endometrial Cells and the Possible Role of LPS in This Process"

_biology, 2024, doi:10.3390/biology13040249_

Round 1

Reviewer 1 Report

Comments and Suggestions for Authors

This paper was a pleasure to read. The experimental design was well conceived, and the methods and results are described clearly. A few minor comments.

L45. Bracket missing after 10

L125. It is mentioned that the fusarium mycotoxin deoxynivalenol (DON) was used as positive control. However the source and other information relating to dose, method of application etc is missing and it is not mentioned in the results.

L152. The RT-qPCR results were less clear cut than for the other methods used. GAPDH was used for normalization but no indication has been provided that the treatments used did not alter GAPDH expression. This should be checked and the information added to the text as this could potentially affect the normalized results that have been reported.

Figures 2 and 4 should be given scale bars.

Author Response

Reviewer 1: Thank you very much for taking the time to review this manuscript. We appreciate the comments and suggestions which helped to improve our manuscript. All questions and comments have been fully answered below.

L45. Bracket missing after 10.  

The suggested correction has been made.

L125. It is mentioned that the fusarium mycotoxin deoxynivalenol (DON) was used as positive control. However the source and other information relating to dose, method of application etc is missing and it is not mentioned in the results. 

We appreciate the comment and suggestion. The information was added. Regarding the results of the cells treated with DON, these are not shown because the purpose of using DON as a positive control was to check the morphological changes at the nuclear level, to observe apoptosis in the cells under study. These changes were not quantified with DON since the negative control was used to compare with the treatments.

L152. The RT-qPCR results were less clear cut than for the other methods used. GAPDH was used for normalization but no indication has been provided that the treatments used did not alter GAPDH expression. This should be checked and the information added to the text as this could potentially affect the normalized results that have been reported. 

Thank you for pointing this out. The results of the expression of the GAPDH gene were clarified in the manuscript.

Figures 2 and 4 should be given scale bars.

The suggested correction has been made.

Please see the attachment: 
Reviewer 1: corresponding revisions/corrections highlighted in red

Reviewer 2 Report

Comments and Suggestions for Authors

Materials and methods - it is important to test the SFB that was used in the culture media to confirm the absence of viral contaminants, particularly for BoHV-4, has this been done? If so, mention it at work.

Results. Figure 2. Include photos of other groups to be able to compare results.

Results. Figure 4. Include photos of other groups to be able to compare results.

Results. Figure 6. show the expression of Bax and Bcl-2 genes for the different treatment groups, as done for caspase-3.

Discussion. Explain a little more about the correlation of work with practical application.

Line 45 - Check Reference Formatting, Missing ] and Space.

Line 64 - Begins paragraph with [8], it would be better to place the authors if it is allowed by the journal's rules, as in Line 154.

Line 111, 124, 139 - used hr to abbreviate time and at other times uses only the h, decide on one, preferably h and use it in all abbreviations.

Line 128 - Remove Spacing Before 2-Phenylindole

Line 133 and 274 - space before citation [19] and [28]

Line 148 - used abbreviation hpi (hours post-infection) and did not write it in full before abbreviating so that the reader knows what it means when it appears this abbreviation again, did this only on Line 209, it is recommended to do this the first time the abbreviation happens.

Line 148 and 151 - unit of temperature away from the number, remove space. Ex: 80°C.

Line 216 - Remove space between before the word magnification.

Line 280 - remove space between before the word is.

Line 287 - Remove space between before the word presence.

Line 288 - remove space between before the word BoHV-4.

Line 336 - Remove space between before the word in MDBK.

Line 355 - Remove space between before the word in early.

Line 376, 378, 386, 390, 393, 395, 397 and 406 - check space between number and first author.

Author Response

Reviewer 2: Thank you very much for taking the time to review this manuscript. We appreciate the comments and suggestions which helped to improve our manuscript. All questions and comments have been fully answered below.

Materials and methods - it is important to test the SFB that was used in the culture media to confirm the absence of viral contaminants, particularly for BoHV-4, has this been done? If so, mention it at work. 

Thank you for pointing this out, this information can be found in the manuscript: "The BEC cells obtained for primary culture and the fetal bovine serum (FBS) used for their growth were tested by virus isolation in MDBK cells and antigen detection by direct immunofluorescence (DIF) using FITC-labeled polyclonal antibodies (IBR/BHV-1, CJ-F-IBR-10ML; BVDV, CJ-F-BVD-10ML, VMRD, Pullman, WA), as previously described [21]. This test together with nucleic acid detection by PCR or nested RT-PCR [21-23] were conducted to rule out contamination in the starting material with bovine alphaherpesvirus 1 (BoHV-1), bovine gammaherpesvirus (BoHV-4), and bovine viral diarrhea virus (BVDV)."

Results. Figure 2. Include photos of other groups to be able to compare results.

The suggested photo has been include.

Results. Figure 4. Include photos of other groups to be able to compare results.

As suggested by the reviewer, we have included a  photo of the control cells.

Results. Figure 6. show the expression of Bax and Bcl-2 genes for the different treatment groups, as done for caspase-3.

The effect of stimulation with LPS was not evaluated in the expression of Bax and Bcl-2 because in the first instance we evaluated the balance of expression of anti-/pro-apoptotic genes induced only by viral infection in the presence of BoGHV4.

Discussion. Explain a little more about the correlation of work with practical application.

The application of the results obtained was developed in the final conclusion.

Line 45 - Check Reference Formatting, Missing ] and Space. 

Line 64 - Begins paragraph with [8], it would be better to place the authors if it is allowed by the journal's rules, as in Line 154.

Line 111, 124, 139 - used hr to abbreviate time and at other times uses only the h, decide on one, preferably h and use it in all abbreviations.

Line 128 - Remove Spacing Before 2-Phenylindole

Line 133 and 274 - space before citation [19] and [28]

Line 148 - used abbreviation hpi (hours post-infection) and did not write it in full before abbreviating so that the reader knows what it means when it appears this abbreviation again, did this only on Line 209, it is recommended to do this the first time the abbreviation happens.

Line 148 and 151 - unit of temperature away from the number, remove space. Ex: 80°C.

Line 216 - Remove space between before the word magnification.

Line 280 - remove space between before the word is.

Line 287 - Remove space between before the word presence.

Line 288 - remove space between before the word BoHV-4.

Line 336 - Remove space between before the word in MDBK.

Line 355 - Remove space between before the word in early.

Line 376, 378, 386, 390, 393, 395, 397 and 406 - check space between number and first author. 

All points above were corrected as suggested.

Please see the attachment: 
Reviewer 2: corresponding revisions/corrections highlighted in green

Reviewer 3 Report

Comments and Suggestions for Authors

The article entitled : "Modulation of apoptosis by Bovine herpesvirus type 4 infection in bovine endometrial cells and the possible role of LPS in this process" investigate the relation of pathogenicity 275 of the Argentine strain 07/435 of BoHV-4 with the apoptosis evidenced in culture of BEC cells and the role of LPS in this process. The main gola of work was to provides insight into the factors that may influence the pathogenesis of BoHV-4 to better understand the mechanisms involved in viral infection in bovine. In reviewer opinion, although many techniques were as well as many results were obtained to achieve the aim of study, the article remains to transparent, clear, well designed. The discussion is quite short but essential.

However the reviewer have some compliance to the language and editing of the text.

Please find below detailed list of comments:

line 45 missing bracket in citation

line 64 reference number in the beginning of the sentence?

line 70-71 I suggest adding reference to the statement

line 204 please add reference: in witch literature?

line 216 delate unnecessary gap (space), please check all the text and edit extra gaps, the same comment to lines 280, 287, 288

line 271-274 more suitable to introduction

line 325 please provide authors name, reference number should go at the end of sentence, the same comment to lines 336 and 339

linę 332 affirm or confirm? 

Author Response

Reviewer 3: Thank you very much for taking the time to review this manuscript. We appreciate the comments and suggestions which helped to improve our manuscript.

line 45 missing bracket in citation

line 64 reference number in the beginning of the sentence?

line 70-71 I suggest adding reference to the statement

line 204 please add reference: in witch literature?

line 216 delate unnecessary gap (space), please check all the text and edit extra gaps, the same comment to lines 280, 287, 288

line 271-274 more suitable to introduction

line 325 please provide authors name, reference number should go at the end of sentence, the same comment to lines 336 and 339

linę 332 affirm or confirm? 

All points above were corrected as suggested.

Please see the attachment:
Reviewer 3: corresponding revisions/corrections highlighted in blue
